# NHANES 2011–2014 Reveals Cognition of US Older Adults may Benefit from Better Adaptation to the Mediterranean Diet

**DOI:** 10.3390/nu12071929

**Published:** 2020-06-29

**Authors:** Matthew K. Taylor, Jonathan D. Mahnken, Debra K. Sullivan

**Affiliations:** 1Medical Center Department of Dietetics and Nutrition, University of Kansas, Kansas City, KS 66160, USA; dsulliva@kumc.edu; 2Alzheimer’s Disease Center, University of Kansas, Fairway, KS 66205, USA; jmahnken@kumc.edu; 3Medical Center Department of Biostatistics and Data Science, University of Kansas, Kansas City, KS 66160, USA

**Keywords:** older adults, Alzheimer’s disease, Mediterranean diet, cognition, NHANES

## Abstract

Although the Mediterranean diet (MedD) has gained interest for potential Alzheimer’s disease (AD) prevention, it is unknown how well US older adults follow a MedD. We used two National Health and Nutrition Examination Survey (NHANES) cycles (2011–2014) to conduct our primary aim of reporting population estimates of MedD adherence among older adults (60+ years) in the US (*n* = 3068). The mean MedD adherence score for US older adults was 5.3 ± 2.1 (maximum possible = 18), indicating that older adults in the US do not adhere to a MedD. There were various differences in MedD scores across demographic characteristics. We also assessed the cross-sectional relationship between MedD adherence and cognitive performance using survey-weighted ordinary least squares regression and binary logistic regression models adjusted for 11 covariates. Compared to the lowest MedD adherence tertile, the highest tertile had a lower odds ratio of low cognitive performance on three of five cognitive measures (*p* < 0.05 for each). Sensitivity analyses within participants without subjective memory complaints over the past year revealed similar results on the same three cognitive measures. We conclude that MedD interventions are a departure from usual dietary intake of older adults in the US and are a reasonable approach for AD prevention trials.

## 1. Introduction

It is well documented that the general US population does not consume a high-quality diet [1,2]. In contrast, it is also well known that healthy diets are important for mitigation of disease risk across the lifespan. Within the older adult population, it is speculated that nutrition may play a role in reducing risk of age-related cognitive decline and onset of Alzheimer’s disease (AD) and other dementias [3,4]. AD is a growing concern domestically and across the globe with projections of nearly three-fold increased prevalence in the US by 2050 [5]. At this time, treatments for AD are inadequate [6], therefore the identification of preventive behaviors, including nutrition, is a high priority.

The consumption of a Mediterranean diet (MedD), a diet rich in fruits and vegetables, whole grains, legumes, nuts, fish, and olive oil, was adopted by the USDA as a recommended dietary pattern for the 2015–2020 dietary guidelines for Americans [7] and has been proposed as a potential approach to reduce the risk of cognitive decline in older adults. Supporting this proposed approach are a host of cross-sectional and longitudinal data that demonstrate diets that most resemble a MedD are related to various measures that are interpreted as reduced risk for AD [8]. In addition to these observational data, one randomized clinical trial (RCT) of the MedD within a Spanish cohort (PREDIMED) demonstrated that the MedD improved cognition in older adults compared to a low-fat diet [9,10,11]. Two other trials, one in Australia [12] and the other in the UK [13], found no effect and an opposite effect of the MedD, respectively. Research trials are ongoing; however, no RCTs of the MedD and cognition or AD prevention have been reported in the US.

While it is likely that diet’s potential role in AD prevention begins much earlier in life, at this time AD prevention research (observational and clinical trials) primarily involves cognitively normal older adults. It can be speculated that older adults living in the US do not have high adherence to a MedD pattern, but the magnitude which individuals adhere or not to the MedD is unknown. Considering the interest in the MedD as a potential intervention to reduce risk for cognitive decline and that these RCTs are currently aimed at the older adult population, information with respect to the extent MedD interventions deviate from the current dietary intake of US older adults is valuable.

The primary purpose of this study was to report the estimated adherence to the MedD pattern within the US older adult population using a generalizable, validated MedD index and, secondarily, assess the relationship between MedD adherence and cognitive performance using combined data from the National Health and Nutrition Examination Survey (NHANES) from 2011–2012 and 2013–2014.

## 2. Materials and Methods

### 2.1. Study Population

This study was conducted using data from the publicly available, cross-sectional NHANES survey from 2011–2014. These data cycles were selected due to the availability of comprehensive cognitive evaluation that began with the 2011–2012 cycle and is most recently reported in the 2013–2014 cycle. NHANES data and the variable codebook described are freely available at https://wwwn.cdc.gov/nchs/nhanes/default.aspx. NHANES used a multistage probability sampling design to produce a weighted, representative sample of the US population [14]. The National Center for Health Statistics Research Ethics Review Board approved all NHANES protocols, and all participants gave informed consent. Our sample included adults ≥60 years old that completed two 24-h food intake recalls and completed at least one cognitive assessment (*n* = 3068). Figure 1 illustrates the flow of participants selected for inclusion in this analysis. Reported energy intake from all participants was plausible for a given day. Modified z-scores of energy intake were also calculated and indicated that no extreme outliers existed (no absolute values ≥3.5) [15].

### 2.2. Dietary Intake Assessment

Trained 24-h food recall surveyors conducted two multiple pass 24-h food recalls with participants. The first was a visually-assisted recall conducted in-person at the NHANES Mobile Evaluation Center (MEC). The second 24-h recall interview was conducted over the telephone three to ten days after collection of the first food recall. Participants without a telephone were given a toll-free number to call in order to complete the second 24-h recall. Nutrient and individual food data were quantified using the USDA’s Food and Nutrient Database for Dietary Studies (15). Individual food data were further combined into 37 food and beverage groupings known as the Food Patterns Equivalents Database (FPED) (16). The NHANES FPED datasets are publicly available at the USDA’s Agricultural Research Service website (available at: https://www.ars.usda.gov/northeast-area/beltsville-md-bhnrc/beltsville-human-nutrition-research-center/food-surveys-research-group/docs/fped-databases/). All dietary intake data from both 24-h recalls were aggregated as average intake from the two days for each participant.

### 2.3. Mediterranean Diet Adherence Assessment

MedD adherence scores were calculated using the 18-point, literature-based Mediterranean Diet Index constructed by Sofi et al. [16] with slight modification. Sofi adherence scores are derived by an assigned value of “0”, “1”, or “2” across nine food categories, with higher scores indicating better adherence to a MedD pattern. Higher MedD component scores reflect more intake of each food component except red meat, which receives a higher score due to less intake, and alcohol, which receives a score of 0 for >24 g, 1 for <12 g, and 2 for >12 g to 24 g of intake per day. These scores are calculated using gram intake references for each category, with the exception of a categorical reference for olive oil of 0 = never, 1 = sometimes, and 2 = frequently. MedD scores were calculated using FPED groupings and their reported intakes in grams. FPED groups relevant for calculation of MedD scores in this study are reported in Table 1. Of interest, FPED fruit and vegetable variables were reported as cup equivalent (CE) intakes. Because these groups were reports of aggregated intake based on individual food intakes without knowing the particular fruits and vegetables consumed to arrive at these CE values, we were unable to convert CEs into grams. Thus, we modified the MedD fruit scores to reflect 0 = <1 CE, 1 = ≥1 CE, and 2 = ≥2 CEs and MedD vegetable scores to reflect 0 = <0.5 CEs, 1 = ≥0.5 CEs, and 2 = ≥1 CE per day. We also extracted olive oil intake in grams from the NHANES individual food component intake data and modified the score to reflect 0 = <14 g, 1 = ≥14 g, and 2 = ≥28 g per day (0, 1, or 2 tablespoons, respectively). All other MedD component scores were calculated in accordance with Sofi et al.

### 2.4. Cognition Assessment

A cognitive assessment battery was administered to NHANES survey participants aged ≥60 years that did not require a proxy informant and could read and understand English, Spanish, Korean, Vietnamese, traditional or simplified Mandarin, or Cantonese. The cognitive battery consisted of the word learning and recall modules from the Consortium to Establish a Registry for Alzheimer’s Disease (CERAD), the Animal Fluency Test (AFT), and the Digit Symbol Substitution Test (DSST), which are described in the following sections. Non-response to cognitive testing for any reason was treated as missing data and not included in the analyses.

#### 2.4.1. Consortium to Establish a Registry for Alzheimer’s Disease (CERAD)

The CERAD assesses immediate learning and delayed recall of new verbal information [17]. Three consecutive word learning trials of 10 words were administered. Participants were presented with 10 of the same unrelated words printed in a different order at each trial. After reading the 10 words out loud, participants immediately recalled as many of the presented words as possible. The total number of correct immediately recalled words was calculated as the individual trial word-learning score. The CERAD immediate learning score for this analysis was an average number of words recalled across the three trials. Delayed recall was scored by asking participants to recall as many words as possible from the originally presented list of 10 after administration of AFT and DSST (approximately 8-10 min after original presentation of the word list).

#### 2.4.2. Animal Fluency Test (AFT)

The AFT assesses the categorical verbal fluency domain of executive function [18]. The AFT asks that participants name as many animals as possible in one minute. One point was assigned for each animal named within the timed portion of the test. In order to acclimatize to the categorical naming demand of the AFT, NHANES participants were administered a practice test that required they name three clothing items. If the participant was unable to name three clothing items, the AFT was not administered.

#### 2.4.3. Digit Symbol Substitution Test

The DSST is a subtest within the Wechsler Adult Intelligence Scale (WAIS III) that assesses processing speed, sustained attention, and working memory [19,20]. In a legend located at the top of the paper test, nine numbers are paired with unique symbols. NHANES participants had 2 min to draw the unique symbol that corresponded with the number into 133 paired, blank boxes. Scoring for the DSST is the total number of correct symbol and number pairs in the allotted time. A standardized practice test was administered prior to initiation of the scored portion of the DSST. Participants who were unable to match symbols with the numbers without assistance did not complete the DSST.

#### 2.4.4. Calculation of Education-Dependent Cognitive Z-Scores

Because education level is a significant contributor to cognitive performance, we calculated education-dependent z-scores for each participant. Individuals were stratified by five education levels, <9th grade, 9–11th grade, high school graduate or GED, some college or associate’s degree, and college graduate or above. Education-dependent scores for each cognitive test were centered and scaled to have mean of 0 and standard deviation of 1 within each education strata. A global cognitive measure was calculated as the average of standardized scores from each individual cognitive test. Individual and global standardized cognitive scores <−1 were characterized as “low cognitive performance” for their respective cognitive measure.

### 2.5. Subjective Memory Changes

NHANES participants were asked medical condition interview questions in the home by trained interviewers. Males and females aged ≥60 years were asked, “During the past 12 months, have you experienced confusion or memory loss that is happening more often or is getting worse?”. Answers to this question were used to stratify participants into two groups, those reporting subjective memory changes and those with no subjective memory changes.

### 2.6. Covariates

Study covariates included age as a continuous variable, sex, BMI as a continuous variable, race/ethnicity, ratio of family income to poverty level as a continuous variable (reported family income divided by the Health and Human Services poverty guidelines specific to the survey year), marital status, smoking status as a categorical variable, diabetes status as a categorical variable, history of cardiovascular disease as a categorical variable, history of hypertension as a categorical variable, and history of stroke as a categorical variable. Covariates were not missing data except for education (*n* = 4), family income to poverty level (*n* = 246), and marital status (*n* = 3). Hours of moderate-vigorous exercise per week was considered as a covariate, but 60% of reported data was missing and not used in the analyses.

### 2.7. Outcomes

The primary aim of this study was to report the mean population adherence to a MedD in adults ≥60 years old and explore categorical differences in adherence to this particular diet pattern. The secondary aim was to assess the cross-sectional relationship between MedD adherence scores and five indicators of cognition (CERAD Immediate Learning, DSST, AFT, CERAD Delayed Recall, and global cognition).

### 2.8. Statistical Analyses

To account for complex survey design and produce representative estimates of the US population, analyses were conducted using the [survey] package for R (v. 3.6.1; R Foundation, Vienna, Austria). Four-year survey weights were calculated and used in all analyses to adjust for unequal selection probability and non-response bias in accordance with NHANES analytical guidelines [21]. Population means, proportions, and standard deviations were estimated and reported. Unless otherwise indicated, statistical analyses were adjusted for all covariates. Observations with missing covariate data were excluded from cognitive statistical analyses. MedD scores and cognitive scores were treated as continuous measures and modeled using survey-weighted ordinary least squares (OLS) regression to assess the linear relationship between MedD scores and education-dependent, standardized cognitive scores. We conducted trend analyses using kernel smoothing to explore these relationships. We observed a sharp upward trend in cognitive test performance in the top third of MedD scores. Thus, tertiles with equal group samples were calculated to characterize “lowest”, “middle”, and “highest” MedD adherence categories to further examine these relationships. We divided education-dependent, standardized cognitive scores into binary variables (“low” < −1 and “not low” ≥ −1) and conducted survey-weighted binary logistic regression to calculate odds ratios (OR) for low cognition across the three MedD adherence tertiles. Assumptions of OLS models were evaluated through residual analyses (e.g., quantile-quantile plots and residual histograms). All other categorical assessments by MedD adherence tertile were performed by survey-weighted Pearson’s chi-squared tests and mean differences of continuous variables were performed using survey-weighted ANOVA using Tukey’s HSD adjustment for multiple comparisons, a special case of OLS regression. We also conducted a sensitivity analysis only with participants that reported no subjective complaints of memory change over the past 12 months. We calculated new MedD tertile scores in this subset and used binary logistic models to assess odds ratios for low cognitive performance across the new MedD adherence tertiles. Statistical significance was set at *p* < 0.05. R statistical code is included as Appendix A.

## 3. Results

Data from 3068 participants aged ≥60 years (mean ± SD: 69.4 ± 11.1 years) were included. Demographic, anthropometric, and dietary intake data are presented in Table 2.

### 3.1. Mediterranean Diet Scores

Mean MedD adherence among older adults from 2011–2014 was 5.3 ± 2.1 on an 18-point scale. Among tertiles of MedD adherence, the mean for the lowest tertile was 3.2 ± 0.9, the middle tertile was 5.4 ± 0.5, and the highest tertile was 7.8 ± 1.3. Across MedD tertile adherence categories, the proportion of non-Hispanic white participants was highest in the lowest tertile and decreased significantly from the lowest to highest tertile while the proportion of non-Hispanic black, Mexican American, other Hispanic, and non-Hispanic Asian participants increased with higher MedD adherence.

Mean estimates of MedD component scores across the population are presented in Table 3. All but alcohol and red meat mean component scores were less than 1 point, with cereals and olive oil components near 0 points. Table 3 also describes the differences in mean component scores across MedD adherence tertiles. Component scores were significantly higher stepwise across MedD adherence tertiles (*p* < 0.01 for all) except for cereals (whole grains), which had similar mean scores for each group.

Categorical differences in mean MedD scores are presented in Figure 2. Females had higher MedD scores than males. Broken into 5-year age categories, older adults aged 70–74.9 years had higher MedD scores (5.7 ± 2.7) than all other age categories and 60-64.9 years had higher scores (5.3 ± 3.1) than those aged 80+ years, the age group with the lowest scores (5.0 ± 1.9). Among race/ethnicity categories, non-Hispanic white participants had lower MedD scores (5.2 ± 2.1) relative to all other categories other than other race, and non-Hispanic Asian participants had significantly higher scores than all race/ethnicity groups (6.7 ± 2.6). There was little significant difference in MedD scores by education level; however, older adults that completed college or above had significantly higher MedD scores than all other individuals (5.8 ± 2.7). Accounting for income, those reporting an income to poverty ratio >2 had slightly higher MedD scores (5.4 ± 4.2) than those reporting a ratio of 1–2 (5.2 ± 2.0) or <1 (5.0 ± 3.4). Cohabitating, non-married individuals had much lower MedD scores (4.2 ± 3.4) with a wide degree of variation compared to all other categories of marital status. There was no difference in MedD scores between individuals with or without diabetes.

### 3.2. Cognitive Performance and Mediterranean Diet Scores

We first assessed linear relationships between continuous MedD scores and continuous education-dependent, standardized cognitive scores for each cognitive test and the global cognition measure. Presented in Table 4, unadjusted survey-weighted OLS models (Model 1) demonstrated that higher MedD scores correlated with better global cognitive performance and on all cognitive tests (*p* < 0.04 for all) except the AFT. After adjusting for all covariates of interest (Model 2), these relationships remained for performance on the DSST and global cognition (*p* = 0.02 for both).

We further examined these relationships with binary logistic regression models to calculate odds ratios (OR) and 95% confidence intervals for low cognitive performance across tertiles of MedD adherence. Results from logistic regression models adjusted for all covariates are presented in Table 5. Compared to the lowest MedD adherence group, those in the highest tertile of MedD adherence had lower OR (95% CI) of low cognitive performance on the AFT [0.6 (0.4−0.9)], CERAD Delayed Recall test [0.6 (0.4−0.9)], and for global cognition [0.5 (0.3−0.9)].

We next conducted sensitivity analyses only with participants that did not subjectively report changes in memory over the past 12 months (*n* = 2579) in order to account for participants with potential clinical cognitive impairment. Mean MedD adherence for the lowest, middle, and highest tertiles were 3.3 ± 0.9, 5.4 ± 0.5, and 7.8 ± 1.2, respectively. Results from logistic regression models controlled for all covariates across the three MedD adherence levels within the sensitivity analysis are presented in Table 6. Compared to the lowest MedD adherence tertile, the highest MedD adherence tertile had lower OR (95% CI) of low cognitive performance on the CERAD Delayed Recall test [0.6 (0.4–0.9)] and for global cognition [0.5 (0.3–0.9)].

## 4. Discussion

The current study is the first to use a fixed, generalizable MedD index to estimate the level at which older adults in the US adopt a Mediterranean style eating pattern and provides additional observational evidence that higher consumption of this pattern is related to better cognition. Our analysis revealed that older adults in the US sampled from 2011 to 2014 do not eat a diet that resembles a Mediterranean diet pattern. Although scores across the entire cohort were generally low, individuals with even slightly higher MedD scores had better cognitive performance.

This evidence demonstrates that population estimated MedD adherence is low overall and within various demographic categories of older adults living in the US. As a whole, older adults had a MedD score of 5.3 out of a possible 18 (29% of the maximum score). The most remarkably low MedD component scores were cereals (whole grains) and olive oil with mean scores only slightly greater than zero. In order to receive a score of “1” for cereals, an individual must consume approximately 4½ oz. of whole grains. These data indicate that older adults do not meet this standard and likely consume the largest proportion of their grains as refined grains. Also, of the 3068 participants surveyed, only 91 reported consuming olive oil in any quantity during the 24-h food recalls. Very little report of olive oil consumption is of particular interest as its intake has purported benefits for various health parameters, including cognition [22,23,24,25]. The red meat component had a particularly high score of 75% of the maximum, indicating that estimated red meat intake among older adults is relatively low.

There were MedD adherence differences across multiple demographic factors. Females had higher MedD scores than males. Older adults aged 70–74.9 years had higher adherence scores than all other age groups and those aged 80+ years had lower scores than all other groups. Non-Hispanic White elders had lower adherence scores than all other race/ethnicity groups and Non-Hispanic Asians had much higher scores than all other races, although their mean score was only 38% of the maximum MedD score. Education also made a modest impact on MedD scores. MedD scores were relatively similar among all education levels but increased sharply in those with a college degree or above. Though difficult to interpret, older adults that reported to be cohabitating with their significant other had a mean MedD score of 4.2, significantly lower than all other marital status categories. In accordance with diet quality findings from other studies, MedD adherence was lower in individuals with a household income at or below the poverty level [26,27], indicating that consuming a healthy diet is likely influenced by an economic component [28]. An important consideration in interpreting these demographic-related findings is that although these differences are statistically significant, their real-life significance is presumably low–across all groups, MedD adherence is very low and evidence of higher MedD scores by a few tenths of a point or the like is not a real indicator of meaningfully better diet quality.

This is not the first study to investigate NHANES data using a MedD quality index, yet it is the first to characterize MedD pattern adherence using a generalizable, fixed index. Previous reports have used indices that rely upon sample median values to produce a sample-dependent MedD score [29,30] or make assumptions of portion size [31,32] using Panagiotakos et al. [33] to differentiate MedD consumption among the study sample and investigate whether intake differences are related to desired outcomes. While useful, there are multiple issues with these index scoring methods. First, median-based assignment of dietary adherence scores is purely relative to the specific study sample used to derive the scores and is not generalizable to other study samples, which are highly variable. The median-based approach assigns a score of “0” when individual intake is lower than the selected sample’s sex-dependent median intake or “1” when individual intake is higher than the sex-dependent median across nine MedD categories. Exemplified by the present study, if the sample clusters on the lower end of intake, then individuals with intake above the median reflect high adherence scores that are not truly indicative of actual high MedD adherence. The use of a fixed MedD index within this cohort allowed us to calculate adherence scores that are generalizable across cohorts and characterize the truly low MedD adherence of US older adults.

MedD scores were 4.6 points higher from the lowest tertile of adherence to the highest tertile of adherence, which also correlated with better protection of cognitive performance in three of the five cognition measures from this study. This study adds to the growing body of evidence that higher adherence to the MedD is related to better cognition in older adults, specifically tests of working memory, processing speed, and delayed recall. The lowest tertile of MedD adherence achieved a score of 18% of the maximum score of 18 and the highest adherence tertile achieved a score of 43% of the maximum. These observations suggest that benefits of higher diet quality may not be limited to achieving nearly perfect adherence to a MedD, but that small adherence improvements may provide cognitive benefit. Due to low MedD scores across the entire study population, we were unable to assess whether this trend of better cognitive performance with higher MedD adherence scores further extends to those with high or very high MedD adherence. The extent to which changes in dietary adherence to the MedD potentiate cognitive benefit will need to be studied extensively through RCTs.

The mechanisms by which the MedD may influence cognition in older adults remain unclear. In the past, researchers have been interested in the impact of single nutrients on health outcomes including cognitive performance and AD risk [34,35]. The influence of individual MedD-related food components on cognition and AD risk has also been studied. The relatively recent interest in a holistic view of diet, such as the MedD, is due to the likely synergistic effect of food/nutrients on health outcomes [36]. The cumulative effect of the MedD on cardiometabolic health [37] and glucose metabolism status [38], both risk factors for cognitive decline [39], and the nutrient-density of the diet in general may affect cognition outcomes through multiple mechanisms.

These analyses were conducted using data from NHANES, a nationally representative health and nutrition survey in the United States, which has inherent strengths and limitations. The primary strength of this study is a survey sampling design that allows for population-based estimations with a high level of confidence. There are also several limitations to this study. First, the cross-sectional nature of this analysis does not allow for the assessment of causality and we cannot rule out the possibility that lower cognition may influence food decision making, manifesting as lower MedD adherence scores. Dietary intake methodology that assesses two full days of consumption over a several-day period may not accurately reflect how individual participants have eaten over the course of their lives or if the reported intake reflects actual usual dietary intake. Concern about these methods have previously been raised [40]; however, NHANES methods are an optimal approach for assessing population-level dietary intake [41]. We did not perform mixed-effects analyses on the two-24 h recalls in this study, which described MedD adherence over two given days rather than provide an estimate for usual intake. Finally, these data were sampled from 2011 to 2014, which may have been slightly ahead of increased social popularity of the MedD. It is possible that older adults sampled more recently have made dietary adaptations toward a MedD pattern.

In conclusion, older adults living in the US do not follow a MedD eating pattern and these data suggest that consuming a diet that slightly more resembles a MedD pattern may be protective of cognitive performance. It is reasonable to use the MedD as a dietary intervention as it is a major departure from the general diet pattern of older adults in the US. It is projected that AD prevalence will rise in the coming decades, thus it is imperative to conduct RCTs to test whether the MedD is truly protective of cognition and can effectively reduce the risk, hence the prevalence, of AD.

## Figures and Tables

**Figure 1 nutrients-12-01929-f001:**
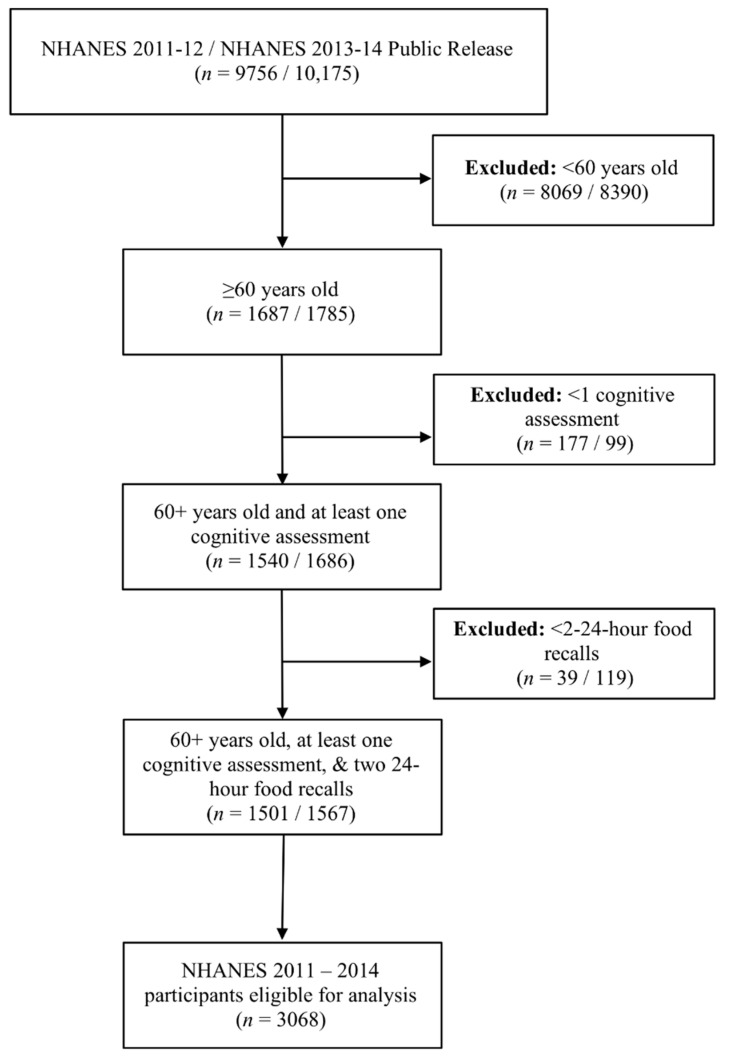
Diagram of the flow of participants included in this analysis.

**Figure 2 nutrients-12-01929-f002:**
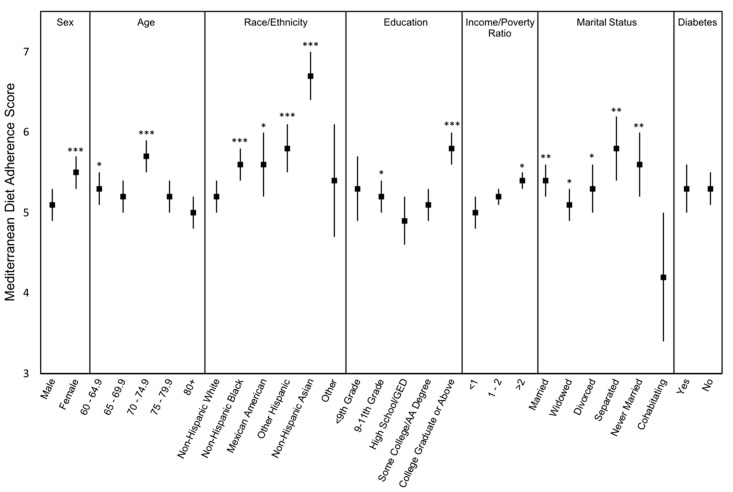
Mediterranean Diet adherence mean scores and 95% confidence intervals of older adults in the US by demographic characteristic. Sex: male (*n* = 1509), female (*n* = 1559); Age: 60–64.9 (*n* = 930), 65–69.9 (*n* = 675), 70–74.9 (*n* = 551), 75–79.9 (*n* = 361), 80+ (*n* = 551); Race: Non-Hispanic White (*n* = 1463), Non-Hispanic Black (*n* = 740), Mexican American (*n* = 273), Other Hispanic (*n* = 301), Non-Hispanic Asian (*n* = 242), Other Race (*n* = 49); Education: <9th Grade (*n* = 412), 9-11th Grade (*n* = 447), High School/GED (*n* = 708), Some College/AA Degree (*n* = 830), College Graduate or Above (*n* = 667); Income to Poverty Ratio: <1 (*n* = 526), 1–2 (*n* = 838), >2 (*n* = 1458); Marital Status: Married (*n* = 1660), Widowed (*n* = 632), Divorced (*n* = 425), Separated (*n* = 82), Never Married (*n* = 187), Cohabitating (*n* = 79); Diabetes Status: Yes (*n* = 741), No (*n* = 2337). Analyses were survey-weighted ordinary least squares regression models using Tukey’s HSD for multiple pairwise comparisons. Models were not adjusted for covariates. *p*-values based on mean difference from the lowest mean value of each demographic category. * *p* < 0.05, ** *p* < 0.01, *** *p* < 0001.

**Table 1 nutrients-12-01929-t001:** FPED intake variables included in Sofi et al. MedD score calculation.

Sofi MedD Score Component	FPED Variable
Fruit	Citrus, Melons, and Berries
	Other Fruit
Vegetables	Dark Green Vegetables
	Tomatoes
	Red/Orange Vegetables
	Other Starchy Vegetables
Legumes	Legumes (As Protein)
Cereals	Whole Grains
Fish	Low Omega-3 Seafood
	High Omega-3 Seafood
Meat	Red Meat
	Cured Meat
	Organ Meats
Dairy	Total Dairy
Alcohol	Alcoholic Drinks
Olive Oil	Olive Oil (grams) ^1^

^1^ Not an FPED variable. Intake was extracted from the individual food data reported in grams of intake. FPED, Food Patterns Equivalents Database; MedD, Mediterranean diet.

**Table 2 nutrients-12-01929-t002:** Baseline Demographics.

		Mediterranean Diet Adherence			
	All	Lowest (*n* = 1023)	Middle (*n* = 1023)	Highest (*n* = 1022)	*p*
Age, y	69.4 ± 6.8 ^1^	69.5 ± 6.8	69.6 ± 6.8	69.1 ± 6.6	0.30
Sex, % female	54.0	47.8	58.3	56.9	0.001
BMI, kg/m^2^	29.1 ± 6.3	29.6 ± 6.6	29.2 ± 6.2	28.3 ± 5.9	0.001
Race/Ethnicity, %					<0.001
Non-Hispanic White	79.0	85.2	78.2	72.4	
Non-Hispanic Black	8.6	6.3	9.1	10.9	
Mexican American	3.5	2.8	3.8	4.0	
Other Hispanic	3.7	2.3	4.5	4.4	
Non-Hispanic Asian	3.3	1.4	2.7	6.3	
Other	1.9	2.0	1.7	2.0	
Education, %					<0.001
<9th Grade	7.2	6.8	7.3	7.3	
9–11th Grade	10.6	11.3	10.5	10.1	
High School/GED	22.1	24.6	25.0	16.0	
Some College/AA Degree	31.0	34.4	28.3	29.8	
College Graduate or Above	29.1	22.9	28.9	36.8	
Marital Status, %					0.06
Married	62.1	60.0	62.2	64.6	
Widowed	17.6	18.8	18.6	15.2	
Divorced	12.2	12.5	11.7	12.3	
Separated	1.2	0.7	1.6	1.4	
Never Married	4.4	4.1	3.6	5.5	
Cohabitating	2.5	3.9	2.3	1.0	
Family Income/Poverty Ratio	3.1 ± 1.6	3.0 ± 1.6	3.1 ± 1.6	3.2 ± 1.6	0.05
Diabetes, % Yes	19.7	20.3	20.3	18.4	0.78
Hypertension, % Yes	59.8	59.3	61.2	58.9	0.75
Cardiovascular Disease, % Yes	18.3	18.9	20.6	15.2	0.05
Stroke, % Yes	7.4	7.7	8.4	6.0	0.36
Smoking Status, % Yes ^2^	50.3	53.5	49.4	47.2	0.22
Moderate-Vigorous Exercise, hr/wk	4.4 ± 4.1	4.5 ± 4.5	4.3 ± 3.9	4.5 ± 3.9	0.72
Mediterranean Diet Score	5.3 ± 2.1	3.2 ± 0.9	5.4 ± 0.5	7.8 ± 1.3	<0.001
Raw Cognitive Scores					
CERAD Immediate Learning	6.5 ± 1.5	6.4 ± 1.5	6.5 ± 1.6	6.7 ± 1.6	0.007
Digit Symbol Substitution Test	52.3 ± 16.7	51.9 ± 16.6	50.9 ± 16.7	54.5 ± 16.7	<0.001
Animal Fluency Test	18.1 ± 5.7	17.9 ± 5.7	17.9 ± 5.7	18.5 ± 5.7	0.24
CERAD Delayed Recall	6.2 ± 2.3	6.0 ± 2.3	6.2 ± 2.3	6.4 ± 2.3	0.006
Global Cognition ^3^	0.0 ± 1.0	-0.1 ± 0.9	0.1 ± 0.9	0.2 ± 1.1	0.002

Group differences were assessed using complex survey-weighted ordinary least squares regression models and Tukey’s HSD for multiple pairwise comparisons. Models were not adjusted for covariates. *p*-values for Race/Ethnicity, Education, and Marital Status were derived by complex survey-weighted Pearson’s chi-squared tests. AA, Associate of Arts; BMI, body mass index; GED, General Education Development. ^1^ Mean ± SD – all such values. ^2^ Smoking status based on having smoked ≥100 cigarettes in life. ^3^ Average of standardized scores from each individual cognitive test.

**Table 3 nutrients-12-01929-t003:** Mean difference in Mediterranean diet component scores by Mediterranean diet score category.

	All (*n* = 3068)	Lowest (*n* = 1023)	Middle (*n* = 1023)	Highest (*n* = 1022)	*p*
Fruit	0.4 ± 0.7	0.2 ± 0.4	0.4 ± 0.6	0.8 ± 0.8	<0.001
Vegetables	0.6 ± 0.7	0.3 ± 0.5	0.6 ± 0.7	1.0 ± 0.7	<0.001
Legumes	0.4 ± 0.8	0.1 ± 0.4	0.4 ± 0.8	0.9 ± 1.0	<0.001
Cereals	0.0 ± 0.2	0.0 ± 0.1	0.0 ± 0.2	0.1 ± 0.3	0.002
Fish	0.5 ± 0.8	0.1 ± 0.4	0.4 ± 0.8	1.0 ± 0.9	<0.001
Red Meat	1.5 ± 0.7	1.2 ± 0.8	1.7 ± 0.6	1.8 ± 0.5	<0.001
Dairy	0.8 ± 0.9	0.4 ± 0.6	0.9 ± 0.9	1.2 ± 0.9	<0.001
Alcohol	1.0 ± 0.4	0.9 ± 0.4	1.0 ± 0.4	1.1 ± 0.5	<0.001
Olive Oil	0.0 ± 0.1	0.0 ± 0.1	0.0 ± 0.1	0.0 ± 0.2	0.01

Component scores were calculated using survey-weighted population estimations. The maximum possible score for each component was 2 points. Analyses were survey-weighted ordinary least squares regression models using Tukey’s HSD for multiple pairwise comparisons. Models were not adjusted for covariates. *p*-values represent comparisons between the lowest and highest MedD adherence tertiles.

**Table 4 nutrients-12-01929-t004:** Survey weight adjusted ordinary least squares regression models assessing the relationship between continuous MedD scores and education-dependent, standardized cognition scores. (*n* = 3068).

	Model 1 ^1^			Model 2 ^2^		
	β	95% CI	*p*	β	95% CI	*p*
CERAD Immediate Learning	0.05	0.00–0.11	0.04	0.03	−0.02–0.09	0.20
Digit Symbol Substitution Test	0.06	0.01–0.11	0.02	0.06	0.01–0.10	0.02
Animal Fluency Test	0.01	−0.04–0.06	0.70	0.04	−0.02–0.10	0.20
CERAD Delayed Recall	0.05	0.01–0.09	0.01	0.04	−0.01–0.08	0.08
Global Cognition	0.06	0.02–0.08	0.004	0.06	0.01–0.08	0.02

^1^ Model 1 is unadjusted, accounting for MedD scores only. Sample sizes for each cognitive test were: Consortium to Establish a Registry for Alzheimer’s Disease–Immediate Learning (*n* = 2857); Digit Symbol Substitution Test (*n* = 2778); Animal Fluency Test (*n* = 2842); Consortium to Establish a Registry for Alzheimer’s Disease–Delayed Recall (*n* = 2855); Global Cognition (*n* = 2711). ^2^ Model 2 is adjusted for all covariates: age, sex, BMI, race/ethnicity, ratio of family income to poverty level, marital status, smoking status, diabetes status, history of cardiovascular disease, history of hypertension, and history of stroke. Sample sizes for cognitive test analyses were reduced from the original sample due to missing covariate data. Sample sizes for each cognitive test were: Consortium to Establish a Registry for Alzheimer’s Disease–Immediate Learning (*n* = 2596); Digit Symbol Substitution Test (*n* = 2533); Animal Fluency Test (*n* = 2583); Consortium to Establish a Registry for Alzheimer’s Disease–Delayed Recall (*n* = 2594); Global Cognition (*n* = 2471).

**Table 5 nutrients-12-01929-t005:** Survey-weighted odds ratios (95% confidence intervals) for low cognitive performance across tertiles of MedD adherence (*n* = 3068).

	Lowest Tertile	Middle Tertile	Highest Tertile
CERAD Immediate Learning	1.0 (Ref)	1.2 (0.9−1.6)	0.8 (0.6−1.0)
Digit Symbol Substitution Test	1.0 (Ref)	1.1 (0.9−1.4)	0.8 (0.6−1.1)
Animal Fluency Test	1.0 (Ref)	0.9 (0.5−1.1)	0.6 (0.5−0.9) *
CERAD Delayed Recall	1.0 (Ref)	0.8 (0.6−1.0)	0.6 (0.4−0.9) *
Global Cognition	1.0 (Ref)	0.8 (0.5−1.3)	0.5 (0.3−0.9) *

Odds ratios calculated by survey-weighted binary logistic regression adjusted for all covariates: age, sex, BMI, race/ethnicity, ratio of family income to poverty level, marital status, smoking status, diabetes status, history of cardiovascular disease, history of hypertension, and history of stroke. The lowest MedD adherence tertile served as the analytical reference (Ref) category. Sample sizes for cognitive test analyses were reduced from the original sample (3068) due to missing cognitive test or covariate data. Analysis sample sizes were: Consortium to Establish a Registry for Alzheimer’s Disease–Immediate Learning (*n* = 2596); Digit Symbol Substitution Test (*n* = 2533); Animal Fluency Test (*n* = 2583); Consortium to Establish a Registry for Alzheimer’s Disease–Delayed Recall (*n* = 2594); Global Cognition (*n* = 2471). * *p* < 0.05.

**Table 6 nutrients-12-01929-t006:** Sensitivity analysis of survey-weighted odds ratios (95% confidence intervals) for low cognitive performance across tertiles of MedD adherence within National Health and Nutrition Examination Survey (NHANES) participants with no subjective memory complaints (*n* = 2579).

	Lowest Tertile	Middle Tertile	Highest Tertile
CERAD Immediate Learning	1.0 (Ref)	1.2 (0.8–1.8)	0.8 (0.6–1.2)
Digit Symbol Substitution Test	1.0 (Ref)	1.2 (0.8–1.5)	0.8 (0.6–1.2)
Animal Fluency Test	1.0 (Ref)	0.7 (0.5–1.0)	0.6 (0.5–1.0) ^†^
CERAD Delayed Recall	1.0 (Ref)	0.8 (0.6–1.0)	0.6 (0.4–0.9) *
Global Cognition	1.0 (Ref)	0.8 (0.4–1.4)	0.5 (0.3–0.9) *

Odds ratios calculated by survey-weighted binary logistic regression adjusted for all covariates: age, sex, BMI, race/ethnicity, ratio of family income to poverty level, marital status, smoking status, diabetes status, history of cardiovascular disease, history of hypertension, and history of stroke. The lowest MedD adherence tertile served as the analytical reference (Ref) category. Sample sizes for cognitive test analyses were reduced from the sensitivity sample (*n* = 2579) due to missing cognitive test or covariate data. Analysis sample sizes were: Consortium to Establish a Registry for Alzheimer’s Disease–Immediate Learning (*n* = 2228); Digit Symbol Substitution Test (*n* = 2188); Animal Fluency Test (*n* = 2220); Consortium to Establish a Registry for Alzheimer’s Disease–Delayed Recall (*n* = 2226); Global Cognition (*n* = 2142). ^†^
*p* = 0.06, * *p* < 0.05.

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
