# Peer review of "NHANES 2011–2014 Reveals Cognition of US Older Adults may Benefit from Better Adaptation to the Mediterranean Diet"

_nutrients, 2020, doi:10.3390/nu12071929_

Round 1
Reviewer 1 Report
I enjoyed reading the manuscript but suggest to take a good look at the title: the word 'although' was confusing me in first instance and only became clear after reading the whole document. However, the title should be attractive. For that reason you might think of something like: NHANES 2011-2014 reveals that cognition of US older adults may benefit from a better adaptation to the Mediterrean diet.
Author Response
I enjoyed reading the manuscript but suggest to take a good look at the title: the word 'although' was confusing me in first instance and only became clear after reading the whole document. However, the title should be attractive. For that reason you might think of something like: NHANES 2011-2014 reveals that cognition of US older adults may benefit from a better adaptation to the Mediterrean diet.
- Thank you for the title suggestion. Our group liked the suggested title and have chosen to rename the paper with your suggestion.
Reviewer 2 Report
1) Is red meat most of the time included in the MedD ?
2) what about herbs and spices with functional compounds ?
3) how is the income/poverty ratio calculated ?
Author Response
1) Is red meat most of the time included in the MedD ?
Thank you for this question. We realize that we did not make the point in the manuscript that higher red meat component scores are reflective of less red meat intake. The following has been added at line 94:
"Higher MedD component scores reflect more intake of each food component except red meat, which receives a higher score due to less intake, and alcohol, which receives a score of 0 for >24g, 1 for <12g, and 2 for >12g to 24g of intake per day."
2) what about herbs and spices with functional compounds?
This is a good question. MedD indices do not include herbs and spices in their calculation. It is likely that individuals in the Mediterranean region consume these functional foods, which could impact health.
Since we do not have data on herbs and spices, we feel that it is out of the scope of the manuscript to speculate. Although we have not discussed this in this manuscript, we believe this is could be a valuable component related to positive health outcomes for those living in the Mediterranean.
3) how is the income/poverty ratio calculated?
The family income to poverty ratio was a variable provided within the NHANES datasets. This variable was calculated by dividing family (or individual) income by the poverty guidelines.
The following was added at line 164:
"...(reported family income divided by the Health and Human Services poverty guidelines specific to the survey year)"